# Bile Acid–Gut Microbiota Axis in Inflammatory Bowel Disease: From Bench to Bedside

**DOI:** 10.3390/nu13093143

**Published:** 2021-09-09

**Authors:** Min Yang, Yu Gu, Lingfeng Li, Tianyu Liu, Xueli Song, Yue Sun, Xiaocang Cao, Bangmao Wang, Kui Jiang, Hailong Cao

**Affiliations:** 1Department of Gastroenterology and Hepatology, Tianjin Institute of Digestive Diseases, Tianjin Key Laboratory of Digestive Diseases, Tianjin Medical University General Hospital, Tianjin 300052, China; yangmin@tmu.edu.cn (M.Y.); guyu@tmu.edu.cn (Y.G.); lilingfeng0419@tmu.edu.cn (L.L.); liutianyu@tmu.edu.cn (T.L.); songxueli@tmu.edu.cn (X.S.); sun_yue@tmu.edu.cn (Y.S.); doccaoxc@163.com (X.C.); mwang02@tmu.edu.cn (B.W.); 2Graduate School of Tianjin Medical University, Tianjin 300070, China

**Keywords:** inflammatory bowel disease, gut microbiota, bile acids, bile acid-activated receptors, therapy

## Abstract

Inflammatory bowel disease (IBD) is a chronic, relapsing inflammatory disorder of the gastrointestinal tract, with increasing prevalence, and its pathogenesis remains unclear. Accumulating evidence suggested that gut microbiota and bile acids play pivotal roles in intestinal homeostasis and inflammation. Patients with IBD exhibit decreased microbial diversity and abnormal microbial composition marked by the depletion of phylum *Firmicutes* (including bacteria involved in bile acid metabolism) and the enrichment of phylum *Proteobacteria*. Dysbiosis leads to blocked bile acid transformation. Thus, the concentration of primary and conjugated bile acids is elevated at the expense of secondary bile acids in IBD. In turn, bile acids could modulate the microbial community. Gut dysbiosis and disturbed bile acids impair the gut barrier and immunity. Several therapies, such as diets, probiotics, prebiotics, engineered bacteria, fecal microbiota transplantation and ursodeoxycholic acid, may alleviate IBD by restoring gut microbiota and bile acids. Thus, the bile acid–gut microbiota axis is closely connected with IBD pathogenesis. Regulation of this axis may be a novel option for treating IBD.

## 1. Introduction

Inflammatory bowel disease (IBD) is a cluster of chronic and relapsing gastrointestinal inflammatory conditions, the dominating subtypes of which are ulcerative colitis (UC) and Crohn’s disease (CD). IBD has aroused wide attention for its increasing incidence and prevalence worldwide [1]. In addition, IBD is prone to recurrence and persistence and patients with IBD have an increased risk of colorectal cancer [2]. As a multifactorial disease, the precise aetiology and pathogenesis of IBD are complex and remain unestablished. Genetic susceptibility, environmental factors (e.g., diet, smoking and antibiotics), gut dysbiosis and host immune response have been reported to be closely associated with IBD pathogenesis [3]. Changes in environmental factors and gut microbiota likely account for the spread of IBD globally [4].

Trillions of microbes inhabited in human gut are involved in various biological metabolisms and immune processes in the host. Gut microbiota has essential roles in maintaining intestinal homeostasis and disturbance of the microbial community emerges as a key factor for the occurrence and progression of IBD [5]. The regulatory functions of gut microbiota are principally based on the abundant microbial metabolites of dietary substrates, such as bile acids and short-chain fatty acids (SCFAs). Bile acids are the metabolic products of dietary substrates; they play a vital role in metabolism and immunological regulation [6,7]. Primary bile acids are synthesized from cholesterol in the liver and then transported to the intestine, where they are converted into secondary bile acids by some specific bacterial species (e.g., *Clostridium* and *Eubacterium*) [8]. Bile acids could bind with bile acid-activated receptors and exert regulatory effects. These receptors include farnesoid X receptor (FXR), Takeda G-protein receptor 5 (TGR5), pregnane X receptor (PXR) and vitamin D receptor (VDR). Abnormal bile acid metabolism has been reported in patients with IBD and interactions between bile acids and gut microbiota have been emphasized in the pathogenesis of IBD [9,10,11,12].

The present review concentrated on discussing the complex interactions between bile acids and gut microbiota in IBD and summarized the potential therapeutic approaches targeting the bile acid–gut microbiota axis for IBD.

## 2. Bile Acid–Gut Microbiota Axis in IBD

Bile acids and gut microbiota have bidirectional effects on each other. Gut microbiota is engaged in the synthesis and transformation of bile acids, (Figure 1) which could affect the microbial composition. Dysregulation of bile acids or bile-acid activated receptors cooperates with gut dysbiosis to cause intestinal inflammation (Figure 2).

Primary BAs (CA and CDCA) are synthesized in the liver and then conjugated to glycine or taurine to form into conjugated BAs (G/TCA and G/TCDCA), respectively. They are stored in the gallbladder and excreted into the intestine to be further metabolized by gut microbiota postprandially. In the terminal ileum, most conjugated BAs are reabsorbed and the others go through the deconjugation mediated by colonic microbiota. In colon, the unconjugated BAs are further transformed into secondary BAs (DCA and LCA) via 7α-dehydroxylation and finally excreted in feces. In IBD, gut dysbiosis affects the metabolism and mainly reduces the deconjugation and 7α-dehydroxylation, leading to the depletion of secondary BAs.

Multiple risk factors (e.g., genetics, western diets and antibiotics) could cause gut dysbiosis and BA perturbations and they are associated with IBD pathogenesis. Under IBD conditions, the microbial diversity is reduced remarkably, with a decreased level of *Firmicutes* and an increased level of *Proteobacteria*. Beneficial bacteria are reduced, whereas pathogens are increased. Most importantly, the abundance of BSH and BAI containing bacteria, such as *Ruminococcaceae*, *Lachnospiraceae* and *Eubacterium* declines. Secondary BAs (such as DCA, LCA and tauro-LCA) are decreased, whilst primary and conjugated BAs (such as CA, CDCA and G/TCA) are elevated because of impaired transformation induced by gut dysbiosis. In turn, disturbed BA profile influences the microbial composition. Moreover, absence or inactivation of BA-activated receptors FXR, TGR5, PXR and VDR disrupts intestinal barrier and incurs bacterial translocation. Afterwards, the dysbiosis could modify colitis-associated gene expression by reprogramming DNA methylation. BAs could be transported by ASBT and OSTα/β receptors or passively diffused across the enterocytes and interact with mucosal immune cells. The disturbed gut microbiota and BA profile, especially the reduced secondary BAs, could influence gut immunity, including breaking the balance between Th17 and Treg cells and increasing ILC3 and ILC1 differentiation. Furthermore, inactivation or absence of FXR and TGR5 in macrophages and dendritic cells increases the production of pro-inflammatory cytokines.

### 2.1. Gut Dysbiosis in IBD

A recent review comprehensively summarized the evidence for the vital role of gut microbiota in IBD, including multiple human and animal studies [13]. Although no specific microbiota is consistently associated with IBD because of different populations, samples and detection methods, the overall trends of gut dysbiosis are consistent. Most studies have revealed reduced microbiome biodiversity in patients with IBD. Meanwhile, the microbial composition is altered, characterized by the reduction in phylum *Firmicutes* and the enrichment of phylum *Proteobacteria*. Beneficial bacteria are decreased, whilst pathogens including *Escherichia coli* are increased. These are the most common microbial characteristics of patients with IBD. But there are some differences between patients with CD and UC. The dysbiosis of CD is much greater than that of UC, with lower diversity and greater changes in composition [14,15,16]. In patients with CD, the abundance of *Erysipelotrichales*, *Bacteroidales* and *Clostridiales*, *Ruminococcaceae* and *Lachnospiraceae* families and *Faecalibacterium prausnitzii* was reduced, whereas that of *Enterobacteriaceae*, such as *E. coli*, *Pasteurellacaea*, *Veillonellaceae*, and *Fusobacteriaceae* families, was elevated [14,17,18,19]. In patients with UC, reductions in *Clostridium* XIVa, *Butyricicoccus*, *Eubacterium rectale*, *F. prausnitzii* and *Roseburia hominis* of the *Ruminococcaceae* and *Lachnospiraceae* families were found, whilst *Ruminococcus gnavus*, *Clostridium ramosum* and *E. coli* were enriched [20,21,22]. Intriguingly, some beneficial bacteria, such as *F. prausnitzii* and *R. hominis*, are SCFA-producing bacteria and they play vital roles in bile acid metabolism.

Gut dysbiosis may participate in the pathogenesis of IBD mainly through the following mechanisms. Ansari et al. [23] found that the gut microbiota under acute inflammatory conditions could modify the host gene expression by reprogramming DNA methylation, leading to upregulation of colitis-associated gene expression (such as AP1, FOSL2 and FRA1). Besides, the dysbiosis in IBD, especially the reduction in SCFA-producing bacteria, leads to decreased production of SCFAs (such as acetate, propionate and butyrate). SCFAs are important energy sources of intestinal epithelium and devoted to strengthening the intestinal barrier function [24]. They also induce the differentiation of regulatory T (Treg) cells and maintain the immune homeostasis [25]. Thus, the reduction of SCFAs in IBD results in impaired barrier and reduced Treg cells. Furthermore, gut microbiota could regulate intestinal immunity. On the one hand, gut microbiota has the property of shaping intestinal innate lymphoid cells (ILCs) [26]. In IBD, macrophages respond to the gut microbiota and lead to the activation of ILC3 and dendritic cells promote the differentiation from ILC3 toward ILC1 [27,28]. The dysregulation of ILC3 and ILC1 leads to elevated production of pro-inflammatory cytokines interleukin-22 (IL-22), IL-17 and interferon-γ (IFN-γ) [29]. On the other hand, gut microbiota is implicated in the modulation of adaptive immunity. The microbiota from patients with IBD could induce intestinal immune disorders in sterile mice, as manifested by increased T helper 17 (Th17) cells with pro-inflammatory cytokines (e.g., IL-17 and IL-22) and reduced Treg cells with anti-inflammatory cytokines, such as IL-10 and transforming growth factor-β (TGF-β) [30,31]. Intriguingly, Devkota et al. found that the pathogen *Bilophila wadsworthia* was associated with colitis in *IL*^10−/−^ mice [32]. *B. wadsworthia* bloomed following the increase in taurine-conjugated bile acids, leading to activated dendritic cells and subsequent Th1-mediated colitis. The present study suggested that the crosstalk of bile acids and gut microbiota may be implicated in the pathogenesis of IBD.

### 2.2. Bile Acid Synthesis and Metabolism

Primary bile acids, which are synthesized in the hepatocytes, consist of cholic acid (CA) and chenodeoxycholic acid (CDCA) in humans. The synthetic process involves a series of complex enzymatic reactions via two pathways initiated by cholesterol 7α-hydroxylase (CYP7A1) and sterol-27-hydroxylase (CYP27A1). Subsequently, primary bile acids are conjugated to glycine or taurine and then excreted into the intestine. In the distal ileum, approximately 95% of these bile acids undergo enterohepatic circulation through apical sodium-dependent bile acid transporter (ASBT) and organic solute transporter subunit α/β (OSTα/β). The remaining bile acids are further metabolized by the colonic microbiota. Conjugated bile acids mainly go through deconjugation, followed by 7α-dehydroxylation and epimerization via gut microbiota, and converted to secondary bile acids, namely deoxycholic acid (DCA), lithocholic acid (LCA) and ursodeoxycholic acid (UDCA). These bile acids are either passively reabsorbed in the colon or excreted in the feces [33] (Figure 1).

### 2.3. Microbial Modulation of Bile Acid Synthesis and Metabolism

Gut microbiota could regulate liver enzymes CYP7A1 and CYP27A1, thereby affecting bile acids [34]. More importantly, gut microbiota participates in the biotransformation of bile acids via microbial enzymes. Conjugated bile acids go through hydrolysis mediated by bile salt hydrolase (BSH) and they are transformed into unconjugated bile acids. BSH has been identified in extensive bacterial genera, including *Lactobacillus*, *Bifidobacterium*, *Clostridium*, *Bacteroides* and *Enterococcus* [35]. Besides, the conversion of secondary bile acids is attributed to limited bacteria with bile acid-inducible enzymes (BAIs), including *Clostridium* cluster XIVa (e.g., *Lachnospiraceae* and *Ruminococcaceae*) and *Eubacterium* in phylum *Firmicutes* [36]. Therefore, bile acid synthesis and metabolism are regulated by gut microbiota to a large extent.

Duboc et al. found that *Clostridium coccoides*, *Clostridium leptum* and *F. prausnitzii* were reduced remarkably in patients with IBD in remission and flare. *Lactobacillus* and *Enterobacteria* (*E. coli* at a species level) were increased in flare. Compared with healthy subjects, patients with IBD, especially active IBD, showed decreased ratios of *F. prausnitzii* and *E. coli*. *Clostridium* and *F. prausnitzii* are well known to support deconjugation and transformation activities. In the present study, the gut microbiota of patients with IBD also exhibited an impaired ability to deconjugate and transform bile acids. The proportion of conjugated bile acids in feces increased, whilst that of secondary bile acids decreased. In addition, impairment of microbial de-sulphation capability with a high level of 3-OH-sulphated bile acids was reported in IBD [9]. Sinha et al. explored the microbiota and bile acids in UC and familial adenomatous polyposis (FAP) and found that *Ruminococcaceae*, one of the bacteria producing secondary bile acids, was decreased considerably in UC. Stool microbial genomes also showed a decreased expression of BAI genes. Bile acid profile showed remarkably reduced secondary bile acids (LCA and DCA) and increased CDCA in UC. The level of CA also increased but not significant. Furthermore, the microbiota in feces from UC remarkably reduced the production of LCA and DCA in vitro compared with that from FAP. This experiment highlighted the loss of microbial metabolism in UC [10]. Other studies also found that phylum *Firmicutes* (e.g., *Clostridium* and *F. prausnitzii*) was remarkably reduced in patients with IBD [15,16,18,19,37,38,39] (The major changes of gut microbiota and bile acids were listed in Table 1) and genomic analysis hinted the depletion of bile acid biotransformation capabilities and production potential in microbiota in IBD [40,41]. Consistently, the concentration of secondary bile acids (particularly LCA and DCA) decreased substantially, whereas the proportion of primary and conjugated bile acids was elevated. Recently, Yang et al. investigated the differences of bile acids and gut microbiota between patients with UC and healthy controls and analyzed the relationships of bile acids and microbiota. The main conclusions were consistent with previous studies. They found that the secondary bile acids, such as LCA, DCA and tauro-LCA were reduced significantly in UC compared with healthy controls and were positively related to *Faecalibacterium*, *Roseburia*, *Butyricicoccus* and *Clostridium*. The primary and conjugated bile acids (CA, TCA and G/TCDCA) were elevated and were positively correlated with *Enterococcus*, *Klebsiella* and *Streptococcus* [22].

Secondary bile acids, such as DCA and LCA, exhibited anti-inflammatory effect rather than sulphated bile acids. Duboc et al. investigated the effects of different bile acids on epithelial inflammatory response. Primary bile acids (CA and CDCA) showed no effect, whereas secondary bile acids (DCA and LCA) inhibited the IL-8 secretion after IL-1β stimulation in Caco-2 cells. Interestingly, sulphated LCA abolished the effect [9]. In addition, secondary bile acids exhibited anti-inflammatory effects in vivo. DCA and LCA alleviated inflammation in several murine colitis models, as featured by the improvement of weight loss, colon morphology and length, with decreased leukocyte infiltration, disease activity and fecal lipocalin 2 level [10]. Secondary bile acids and their derivatives (e.g., 3-oxoLCA and isoalloLCA) were demonstrated to promote Treg cell differentiation and inhibit Th17 cell differentiation [6,42]. Furthermore, bile acids played critical roles in intestinal innate immunity via bile acid-activated receptors [43]. Interestingly, a recent study found that high-dose and long-term intake of DCA may aggravate intestinal inflammation [44].

### 2.4. Bile Acids Influence the Composition of Gut Microbiota

The bile acids excreted into the intestine are further metabolized by gut microbiota and in turn, bile acids affect the microbial composition. As reported by Islam et al., administration of CA could bring about increased abundance in phylum *Firmicutes*, *such as*
*Clostridium* cluster XIVa, *and* reduced abundance in phylum *Bacteroidetes in rats* [45]. *Another study also found a significant* increase in bile acid 7α-dehydroxylating bacteria in mice supplemented with CA [46]. Bile acids have been demonstrated to exert direct and indirect effects on gut microbiota [47]. On the one hand, bile acids could directly inhibit the growth of bacteria by increasing cell membrane permeability and causing cell damage [48]. They could also induce DNA and oxidative damage in bacteria [47]. On the other hand, bile acids could influence bacteria indirectly through the FXR and VDR [49,50]. Specifically, FXR agonist GW4064 restrained the overgrowth of bacteria induced by biliary duct ligation in mice [50]. Cathelicidin is a major antimicrobial peptide that could curb bacterial growth. CDCA and UDCA induced the expression of cathelicidin through FXR and VDR in vitro. In vivo, UDCA therapy exhibited an ability to increase VDR and cathelicidin expression [49].

### 2.5. Bile Acid-Activated Receptors

Bile acids are natural ligands of several receptors and they are engaged in the regulation of metabolic and immune processes through activating the corresponding receptors [51,52]. Bile acid-activated receptors include FXR, TGR5, PXR and VDR. All of them are nuclear receptors except the membrane receptor TGR5 [43,51].

#### 2.5.1. FXR

FXR is distributed predominately in intestinal epithelial cells, hepatocytes and some immune cells (e.g., macrophages and dendritic cells) in the gut and liver [43]. In enterocytes, the most potent ligand for FXR is CDCA, followed by DCA, LCA and CA [53]. Muricholic acids (MCAs) constitute a part of primary bile acids in mice, of which α-MCA and β-MCA are potent FXR antagonists. The gut microbiota mainly affects FXR through change in bile acids. The antioxidant tempol could induce reduction in *Lactobacillus* and this change could lead to the impairment of BSH activity and subsequent accumulation of tauro-β-MCA and FXR inhibition [54]. Similarly, in patients with IBD, phylum *Firmicutes*, especially the *C. leptum group endowed with BSH activity*, was remarkably reduced [55]. FXR and microbiota have a bidirectional influence. The FXR agonist GW4064 restrains the bacterial overgrowth [50] and thus protects the intestinal tract from bacteria-induced damage. Zhang et al. have reported that the high-affinity FXR antagonist gly-MCA disturbs the gut community structure, as characterized by the reduced ratio of *Firmicutes* to *Bacteroidetes, whilst the application of* GW4064 reverses the alteration [56]. Besides, the synthetic FXR ligand 6-ethyl-CDCA (INT-747) could promote the production of cathelicidin [57]. Therefore, the activation and antagonism of FXR affect the intestinal bacteria.

Some studies showed that FXR is related to the occurrence of IBD. Downregulated expression of FXR mRNA in inflamed colonic mucosa was observed in patients with CD and colitis mice models [58]. Studies pointed that *FXR^−/−^* mice exhibited increased expression of inflammatory cytokines and aggravated 2,4,6-trinitrobenzenesulfonic acid (TNBS) and dextran sulfate sodium (DSS)-induced colitis, whilst FXR activation could alleviate colitis via application of INT-747 [57,58]. FXR was involved in modulating intestinal immunity [43,58] and it may influence IBD through immune activities. Immune perturbations, such as elevated infiltration of macrophages, were observed in *FXR^−/−^* mice. Macrophages isolated from TNBS-treated *FXR^−/−^* mice released more inflammatory cytokines than wild type (WT) mice. Conversely, FXR activation in macrophages inhibited the generation of nuclear factor-κB (NF-κB)-dependent pro-inflammatory cytokines, such as IL-6, IL-1β and tumor necrosis factor-α (TNF-α) [58]. Another study found that the ligand INT-747 reduced TNF-α secretion in monocytes and dendritic cells [57]. The two studies identified that FXR activation alleviated colitis. Moreover, FXR could protect the intestinal barrier in vivo and vitro and decrease goblet cells loss, thus contributing to the inhibition of intestinal inflammation [57,59]. Gadaleta et al. investigated the role of downstream fibroblast growth factor 19 (FGF19) and found a reduced level of FGF19 in patients with CD [60]. In this study, FGF19-M52, a variant of the FGF19 protein was applied and demonstrated to maintain the intestinal barrier, inhibit inflammatory immune response (reduced macrophages recruitment and pro-inflammatory cytokines) and regulate the gut microbial community, leading to alleviation of colitis in a FXR-dependent manner. Xu et al. found that FXR agonist fexaramine could restore the FXR-FGF15 activity and normalize bile acid metabolism in mice. The abundance of SCFA-producing bacteria was elevated. As a result, FXR activation attenuated intestinal inflammation induced by high-dose DCA. Besides, administration of antibiotics also reduced intestinal inflammation. This study hinted the key role of FXR and gut microbiota in intestinal inflammation [44]. Thus, the activation of FXR and its downstream signaling pathway exert protective effects on intestinal inflammation. The mechanism may involve restoration of bile acid metabolism and gut microbiota, as well as positive effects on intestinal immunity and barrier.

#### 2.5.2. TGR5

TGR5 is another major and extensively studied bile acid-activated receptor [61]. Secondary bile acids, especially LCA and DCA, are the most potent ligands binding to TGR5 [53]. In addition, TGR5 participates in the modulation of metabolic process and immune response [51,61].

TGR5 could protect mice from colitis. *TGR5^−/−^* mice presented higher susceptibility to colitis and more severe inflammation than WT mice. Activation of TGR5 could ameliorate intestinal inflammation [62]. Sinha et al. demonstrated that the anti-inflammatory effects of LCA were closely related to TGR5 as the protective effect against colitis was lost in *TGR5^−/−^* mice compared with that in WT mice. They further studied the role of immune cells expressing TGR5 through bone marrow transplantation experiment. Based on the same DSS-induced colitis and LCA administration, the mice who received bone marrow from *TGR5^−/−^* mice were in worse inflammatory condition than those who received it from WT mice. In addition, the former had more TNF-α^+^ and IL-17^+^ colonic leukocytes. The results implied that immune cells expressing TGR5 play an essential role in LCA protecting the host against colitis [10]. TGR5 may influence inflammation through regulating immune activities. Macrophages play significant roles in regulating cytokine production and inflammatory response in the gastrointestinal tract and they could be regulated by TGR5 [63]. Macrophages manifest the plasticity to differentiate into two distinct phenotypes: classically activated macrophages (M1) and alternatively activated macrophages (M2). M1 shows the pro-inflammatory property of producing cytokines IL-12, IL-6, IL-1β and TNF-α, whilst M2 exhibits anti-inflammatory specialty characterized by the production of IL-10 [64,65]. Macrophage polarization under the regulation of TGR5 directly affects the tendency of inflammatory response (pro-inflammatory or anti-inflammatory) [7]. Several studies have found that TGR5 agonists (such as DCA, LCA and tauro-LCA) could suppress the production of TNF-α and IL-12 in lipopolysaccharide-treated macrophages. The IL-10/IL-12 ratio was also elevated, suggesting that macrophages transformed into the anti-inflammatory phenotype. TGR5 could induce cyclic adenosine monophosphate (cAMP) production and subsequent cAMP-dependent protein kinase A(PKA) activation, leading to the inhibition of NF-κB [66,67]. Besides, TGR5 activation was proven to induce monocytes to differentiate into dendritic cells with reduced cytokine IL-12 via the TGR5-cAMP-PKA pathway [68]. Biagioli et al. reported that *TGR5^−/−^* mice presented enhanced M1 recruitment and intestinal inflammation, whereas TGR5 agonist BAR501 promoted the macrophages shifting from M1 to M2 phenotype and brought about reduced pro-inflammatory cytokines and relieved experimental colitis. A recent study found that activation of the TGR5-cAMP-PKA axis also caused ubiquitination of Nod-like receptor protein 3 (NLRP3). As a result, the production of NLRP3 inflammasome and subsequent IL-1β excretion were blocked [69]. In addition to causing immune dysfunction, TGR5 deficiency contributed to the impaired gut mucosal barrier and increased intestinal permeability [62]. Therefore, the activation of TGR5 may relieve colitis by modulating immunity and improving the intestinal barrier function.

#### 2.5.3. PXR

PXR is expressed in the liver and intestine and it could be activated by LCA and its metabolite 3-keto-LCA [51]. As the principal xenobiotic receptor, PXR could achieve detoxification and elimination of xenobiotics. Furthermore, PXR is involved in the regulation of inflammatory response, cell proliferation and migration [70]. PXR activation was proven to alter the structure of gut microbial community and regulate bile acids in a bacteria-dependent manner [71]. A recent study has pointed out that statin alters the diversity and composition of gut microbiota in a PXR-dependent manner [72]. Conversely, gut microbiota could modulate PXR through LCA production and another bacteria-derived metabolite indole 3-propionic acid, which is a tryptophan degradation product and a ligand for PXR [73].

In patients with UC, downregulated PXR and its target genes (mainly cellular detoxification and defense genes) were observed, hinting a possible link between PXR and the pathogenesis of IBD [74]. In addition, genetic variations of PXR were identified to be closely related to an increased risk of IBD [75,76]. Mice with PXR deficiency were more susceptible to colitis and showed more severe inflammation than the control, whereas the activation of PXR by agonists could protect from colitis [77]. Zhou et al. found that PXR activation inhibited NF-κB signaling, whereas deletion of PXR elevated the expression of NF-κB and NF-κB activation reciprocally suppressed the expression of PXR and its target genes [78]. The reciprocal suppression may account for a mechanistic link between PXR and inflammation signaling pathways. Studies showed that PXR may contribute to IBD protection via suppression of the NF-κB pathway. In mice treated with DSS, PXR activation suppressed NF-κB activity and inhibited the production of pro-inflammatory cytokines, leading to improvement of intestinal inflammation [77]. PXR gene ablation resulted in impaired intestinal barrier integrity and elevated expression of Toll-like receptor 4 (TLR4). Furthermore, TLR4 signaling was proven to be an essential causative pathway in intestinal barrier disruption because barrier defects were corrected in *PXR**^−^**^/^**^−^*
*TLR4**^−^**^/^**^−^* mice. The PXR ligand indole 3-propionic acid could also decrease intestinal mucosal permeability and the expression of TNF-α via TLR4 signaling. This study suggested that PXR exerts an anti-inflammatory effect by negatively modulating TLR4 signaling [73].

#### 2.5.4. VDR

VDR is widely expressed in various tissues, activated by 1,25-dihydroxyvitamin D and involved in metabolic and immunologic process modulation. Besides, VDR is activated by LCA and 3-oxo-LCA. It helps detoxify LCA in the liver and intestine by inducing the expression of cytochrome P450 enzyme [79]. Gut microbiota could modulate VDR by altering bile acid metabolism because bile acids serve as ligands and regulators of *VDR* expression [49,79]. Inversely, the human genetic variations of VDR were identified to affect the microbial diversity and metabolism. The microbiota in *VDR^−/−^* mice extensively changed compared with that in WT mice [80].

VDR was identified as a susceptibility gene for IBD [81]. Disturbance of VDR expression and signal were reported in patients with IBD [82]. VDR expression in the colon was inversely correlated with the histological score in IBD and VDR staining showed a lower level in the diseased section than the quiescent segment [83]. This study may indicate the protective effect of VDR on intestinal inflammation. On the basis of the results of VDR expression change in IBD, the intestinal inflammation and corresponding pathological mechanism were further studied in VDR-deficient mice. Research showed that *VDR^−/−^* mice developed more severe colitis after TNBS treatment, along with increased intestinal epithelial cell apoptosis and mucosal barrier permeability. The resulting bacterial invasion could lead to immune system disorder, as characterized by the unbalanced Th1/Th17 response. Immune dysregulation and inflammation could be corrected by the depletion of bacteria with antibiotics, suggesting that VDR worked in a bacteria-dependent manner [84]. In a study carried out by Liu et al., epithelial VDR signaling inhibited epithelium apoptosis by suppressing NF-κB signal and the key mediator of apoptosis, the p53-up-regulated modulator of apoptosis. As a result, overexpression of VDR in mice could maintain epithelial barrier function and attenuate colitis [82]. A recently published study showed an indispensable role of VDR in maintaining colonic Treg cell homeostasis, which was critical to resistance to DSS-induced colitis [6]. Besides, Paneth cells could sense gut bacteria through MyD88-dependent TLR activation, inducing the production of antimicrobial peptides [85]. The absence of VDR downregulated autophagy-related 16-like 1 and Beclin-1 resulted in deficits in autophagy [86] and impairment of Paneth cell function [87]. Abnormal Paneth cells and reduced lysozyme may account for gut dysbiosis in mutant mice, a crucial factor in intestinal inflammation [87]. Therefore, VDR may act as a modulator of colitis by regulating autophagy and Paneth cells and further altering the gut community. Meanwhile, administration of microbiota-derived butyrate elevated the expression of VDR and relieved inflammation in DSS-treated mice, thus showing the bidirectional effects between gut microbiota and VDR in intestinal inflammation.

## 3. Therapeutic Target of Bile Acid–Gut Microbiota Axis for IBD

On the basis of the changes and effects of gut microbiota and bile acids in IBD, several therapies that target the bile acid–gut microbiota axis were summarized. The relevant studies are listed in Table 2.

### 3.1. Dietary Therapy

Epidemiologic data have shown the association between diets and risk of IBD [92]. Diet could rapidly and remarkably alter the human gut microbiome and bile acid pool [93,94]. Specific diet-induced alteration of bile acids and microbiota could activate immune cells [32] and increase intestinal permeability [95], leading to the occurrence of intestinal inflammation in mice. Thus, improving IBD by adjusting diets, resulting in the modification of the bile acid–gut microbiota axis, is reasonable. Wang et al. found that a hydrolyzed protein diet could induce relief of chronic inflammation in a canine model. The relief was linked with the increased levels of LCA and DCA and restored gut microbiota marked by reduced pathogens and increased bile acid-producing *Clostridium hiranonis* [89]. Other dietary substances, such as fucose and total alkaloids of Sophora alopecuroides L., also presented some therapeutic effects on colitis in mice [90,91]. They could restore the gut dysbiosis and increase the ratio of *Firmicutes*/*Bacteroidetes*. Besides, the levels of primary and conjugated bile acids declined after treatment and they were similar to those of the control group. Exclusive enteral nutrition (EEN) is a widely studied and established dietary therapy mainly applied to induce remission of pediatric CD [96,97]. EEN may exert anti-inflammatory effects through modulating microbiota, bile acid metabolism and immune activities [98,99]. Studies have found that EEN exhibited effects on the composition and function of microbiota during the treatment of pediatric CD [100,101]. Furthermore, EEN partially restored the abnormal composition of bile acids in patients with CD and brought about increased LCA with reduced primary and conjugated bile acids [39]. Intriguingly, patients with different bile acid profiles and microbial communities have distinct responses to EEN. Patients with primary bile acid as the dominant bile acid showed non-sustained remission or relapse after EEN therapy. They exhibited decreased gut microbial diversity, accompanied by decreased abundance of *Ruminococcaceae* and *Lachnospiraceae*, and increased abundance of phylum *Proteobacteria* [102]. By contrast, studies in adults have exhibited variable results, possibly due to poor adherence to the treatment [103]. Overall, dietary therapy is likely to be a relatively safe approach. However, studies assessing the effects of diverse diets in IBD are difficult owing to various confounding factors and the compelling evidence at present are insufficient. More randomized controlled trials (RCTs) determining the efficacy and further mechanism of specific diet in pediatric and adult populations are warranted. Besides, in view of the challenges of long-term adherence to most existing diet therapies, more accessible nutritional schemes should be explored.

### 3.2. Probiotics and Prebiotics

Probiotic therapy is a method of introducing specific bacteria with well-recognized benefits to competitively inhibit pathogens and normalize the composition of gut microbiota [104]. The most widely used probiotics are *Bifidobacterium* and *Lactobacillus* genera, as well as *Lactococcus* spp., *E. coli* Nissle 1917 and *Streptococcus thermophilus* [105]. Probiotics have been shown to be effective in improving IBD in several clinical studies. *E. coli* Nissle 1917 is a well-studied strain that exhibits a comparable efficacy and safety of maintaining remission for UC with mesalazine [106,107,108]. Other species of probiotics such as *Bifidobacterium* and *Lactobacillus* were also effective in patients with UC [109,110,111]. Compared with the placebo group, probiotics brought about more significant endoscopic and histopathological improvement and lower clinical activity index [109,111]. Probiotics were also more effective for prolonging clinical remission in patients with UC [110]. VSL#3 is a *mixture* of 8 strains of probiotics, consisting of 4 *Lactobacillus* strains, 3 *Bifidobacterium* strains and 1 *Streptococcus* strain. It has been shown that VSL#3 is effective in inducing and maintaining remission in patients with UC [112,113]. As an adjunct to standard therapy, VSL#3 presented better efficacy than single standard therapy both in adults [112] and children [113] *with UC*. The disease activity index was significantly lower in VSL#3 group compared to the placebo group. In patients with UC in the presence of 5-aminosalicylic acid intolerance, the use of VSL#3 was even more important [114]. Probiotics such as VSL#3 and *E. coli* Nissle 1917 have been recommended by European Society for Clinical Nutrition and Metabolism for the treatment of mild or moderate UC [96]. Probiotics tend to restore the abundance of protective bacteria, enhance the intestinal epithelial barrier function and regulate immune activity in the host, which may account for the amelioration of gut inflammation [115,116]. Furthermore, probiotics (*Lactobacillus plantarum CCFM8661*, *Lactobacillus reuteri NCIMB 30242 and* VSL#3) have been demonstrated to influence bile acid metabolism via FGF19/FGF15 in humans/mice, such as promoting the synthesis and excretion of bile acids [117,118,119]. VSL#3 increased the abundance of *Bifidobacteriaceae* and *Lactobacillaceae* with elevated BSH activity, leading to decreased conjugated/unconjugated BAs [119]. However, probiotics usually exhibit low or no effect on CD [120,121,122].

Based on probiotics, researchers have made further efforts. Unlike traditional single probiotics or simple combination, a recent study has proposed the designed bacterial consortia [123]. The bacterial consortia were designed to interdependently restore the microbial composition and function in patients with IBD. The goal was to provide and replenish key therapeutic functions (e.g., conversion of secondary bile acids, especially DCA and LCA, synthesis of SCFAs and indole, synthesis of antimicrobials). The bacterial consortia were networks of metabolically interdependent strains, consisted of more than 10 human strains, referred to as GUT-103 or GUT-108. They were demonstrated to prevent and treat experimental colitis in mice. They reversed gut dysbiosis and restored functionality. Specifically, they could perform varieties of functions, including but not limited to 7-α-dehydroxylation. Although this attempt is still in the preliminary stage, it is indeed a direction worth further study.

Prebiotics, defined as ‘substrates that are selectively utilized by host microorganisms conferring a health benefit’, could promote the growth and metabolic activity of the beneficial resident bacteria of the host. The most commonly available prebiotics are lactulose, fructo-oligosaccharide, galacto-oligosaccharide and inulin [124]. Previous clinical practice has revealed the beneficial effect of prebiotics (e.g., inulin and oligofructose) in UC treatment [125,126]. Prebiotics are usually used in combination with probiotics, named synbiotics. In addition to improving the growth of probiotics, such as *Bifidobacterium*, *Lachnospiraceae* and *Ruminococcaceae* [127,128], prebiotics could affect the production of microbial metabolites SCFAs and bile acids [126,129]. For example, inulin increased the concentration of fecal DCA and LCA in dogs [129]. Overall, probiotics and prebiotics may regulate gut homeostasis by altering the profiles of bacteria and bile acids and be involved in the remission of intestinal inflammation. However, more studies are needed to determine the clear benefits and underlying mechanism.

### 3.3. Engineered Bacteria

Researchers have recently engineered microbes through synthetic biology tools to achieve targeted treatment. The engineered bacteria have been studied under several disease conditions, such as colitis, cancer and pathogenic infection. However, overall, the therapeutic application of engineered bacteria is still in the preliminary research stage, mainly in animal subjects. The following are several application examples of engineered bacteria in the treatment of colitis. Engineered *Lactococcus lactis* was designed to secrete IL-10 [130] or IL-27 [131] and attenuate colitis. This strain could increase the production of IL-10 and inhibit the production of pro-inflammatory cytokines, such as IL-6, IFN-γ and IL-23. Engineered *E. coli* Nissle 1917 produced trefoil factors (TFFs) that promoted gut barrier function and protected mice from colitis through barrier protection and immune regulation [132]. Similarly, TFF was expressed in engineered *Lactococcus lactis* and presented a protective effect [133]. Therefore, engineered bacteria could be a candidate therapy though the mechanisms of action are not yet fully understood. Campbell et al. have developed an engineered *Bacteroides* strain that produced iso-DCA and further increased colonic Treg cells. Iso-DCA interacted with FXR in dendritic cells to enhance its Treg induction [134]. This study demonstrated that engineered bacteria also have crosstalk with bile acids and are engaged in immune regulation, which may play a role in the relief of colitis.

### 3.4. Fecal Microbiota Transplantation (FMT)

FMT is a manipulation of introducing microorganisms from pre-screened healthy donors into patients to ameliorate gut dysbiosis of recipients by normalizing diversity and the function of microbiota [135]. Over the past few decades, FMT has been successfully used to treat *Clostridioides difficile* infections. The mechanisms involve competition of symbiotic microbiota, restoration of bile acid metabolism and improvement of intestinal barrier via mucosal immunity [136]. Unlike probiotics, FMT brings about long-term engraftment [137]. Furthermore, FMT equipped with a largely distinct scale and content from probiotics due to approximately 10^11^ bacterial cells per gram of wet stool, along with fungi, viruses and archaea [138] is expected to establish a broadened microorganic equilibrium beyond bacteria [139]. Several well-designed RCTs have investigated FMT in patients with mild to moderate UC, with the endpoint of steroid-free clinical remission and endoscopic improvement. FMT presented therapeutic benefits over placebo [140,141,142], suggesting that it is a promising induction therapy for UC remission. Paramsothy et al. have found that the improved microbial community with enrichment of *Roseburia inulivorans* and *Eubacterium hallii* and the increased levels of SCFAs and secondary bile acid (dehydrolithocholate) after FMT were responsible for the induction of remission [88]. By contrast, RCTs in CD are limited. Sokol et al. recently performed the first RCT to evaluate FMT in patients with CD. In this study, patients treated with FMT showed significant improvement in endoscopic result and C-reactive protein level compared with the control. The clinical remission rate was higher with FMT, whereas no significant difference was found in the control, which may be attributed to insufficient subjects [143]. An earlier uncontrolled cohort study also reported effective clinical remission but lacked the evaluation of endoscopic remission [144]. These trials revealed that FMT is a potential approach to CD remission. However, RCTs involving a large scale of participants should be carried out to explore the effects of FMT on patients with CD.

Taken together, FMT is a promising therapy for IBD remission. However, the current studies presented variable and inconsistent results in the efficacy of remission, possibly because of the different experimental designs, such as donor selection, application timing and dosage, delivery approaches and efficacy evaluation [145]. Most importantly, safety problem is not negligible due to a reported death incident induced by antibiotic-resistant bacterial infection after FMT [146]. Thus, establishing safe and stable therapeutic protocols should be the priority in clinical practice.

### 3.5. UDCA

UDCA is a secondary bile acid derived from CDCA and it presents at a low concentration in humans [147]. It has well-established therapeutic properties and it is originally used to treat cholestatic liver diseases [148]. UDCA could also improve colitogenic dysbiosis. In particular, the ratio of *Firmicutes/Bacteroidetes* was normalized and the abundance of *Clostridium* cluster XIVa and *Akkermansia muciniphila* was increased. As a result, colitis was relieved in mice [12]. UDCA could upregulate the expression of FXR and VDR and restore bile acid homeostasis [149]. Furthermore, UDCA could reduce the production of pro-inflammatory cytokines [150], inhibit enterocyte death and protect intestinal epithelial barrier integrity [151]. The above effects are remarkably associated with the suppression of intestinal inflammation. However, the clinical practice of UDCA therapy in IBD remission is rare thus far.

Patients with IBD have a high risk of developing colorectal dysplasia and cancer [2]. Clinical trials have shown that UDCA could reduce the risks in patients with UC [152]. The abundances of *F. prausnitzii* and *Ruminococcus gnavus* were increased and decreased, respectively, by UDCA treatment and this finding is associated with the lower risk of colorectal adenoma in men than in women [153]. However, a study has suggested that long-term administration of high-dose UDCA may increase the risk [154]. Therefore, the clinical use of UDCA for patients with IBD, especially in terms of dosage and duration, still needs to be prudent.

Primary sclerosing cholangitis (PSC) is usually linked with IBD, especially UC [155]. Up to 66% of patients with PSC have coexistent IBD (UC accounts for 75%). Patients with PSC-IBD have remarkably different microbiota profiles and impaired bile acid metabolism compared with healthy controls. The bile-acid signaling pathways are also upregulated [156]. UDCA improves serum liver biochemistry and histology index but shows no long-term benefit. Thus, it remains controversial in the treatment of PSC [157].

## 4. Conclusions

Accumulating evidence has recently shown the link between bile acids and gut microbiota and their roles in IBD have caught much attention. However, the intricate interactions between the two in IBD need further elaboration. Gut microbiota is implicated in bile acid metabolism and it affects the composition of bile acids. Inversely, the altered bile acid pool could further disturb microbial homeostasis. Thus, gut dysbiosis, abnormal bile acid profile, and bile acid-activated receptors synergically contribute to IBD development. In this study, several therapies for IBD targeting the bile acid-gut microbiota axis were summarized. In the future, the recovery of bacterial function may be an important direction of treatment of IBD, including further research on probiotics, standardization of FMT, individualized microbial therapy, development of engineered bacteria and bacterial consortia and so on. These therapies appeared to be promising candidate treatments but the clinical efficacy and mechanisms still need to be confirmed by further studies.

## Figures and Tables

**Figure 1 nutrients-13-03143-f001:**
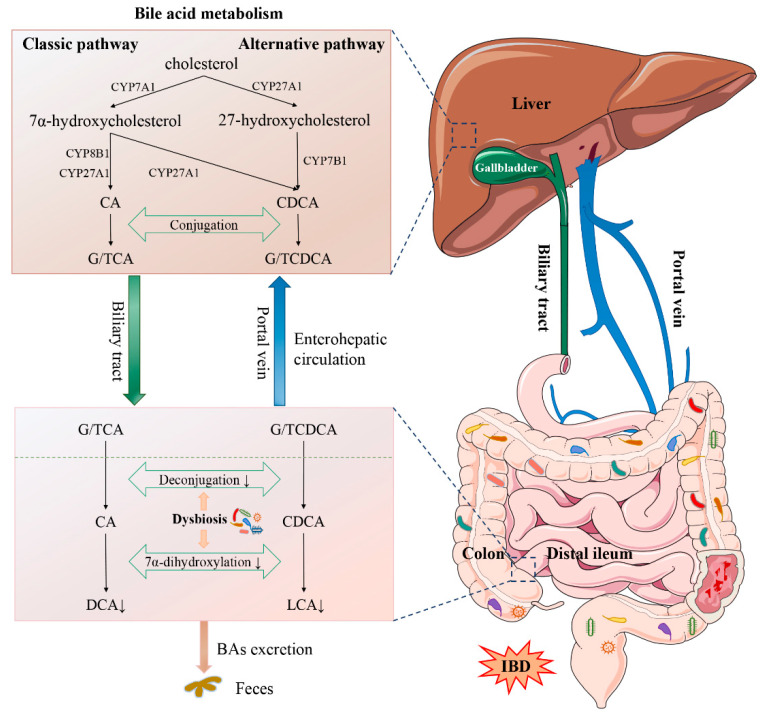
Bile acid synthesis and metabolism and the major changes in IBD. Abbreviations: IBD, inflammatory bowel disease; CA, cholic acid; CDCA, chenodeoxycholic acid; GCA, glycocholic acid; TCA, taurocholic acid; GCDCA, glycochenodeoxycholic acid; TCDCA, taurochenodeoxycholic acid; DCA, deoxycholic acid; LCA, lithocholic acid; BA, bile acid; CYP7A1, cholesterol-7α-hydroxylase; CYP27A1, mitochondrial sterol-27-hydroxylase; CYP8B1, sterol-12α-hydroxylase; CYP7B1, oxysterol 7α-hydroxylase.

**Figure 2 nutrients-13-03143-f002:**
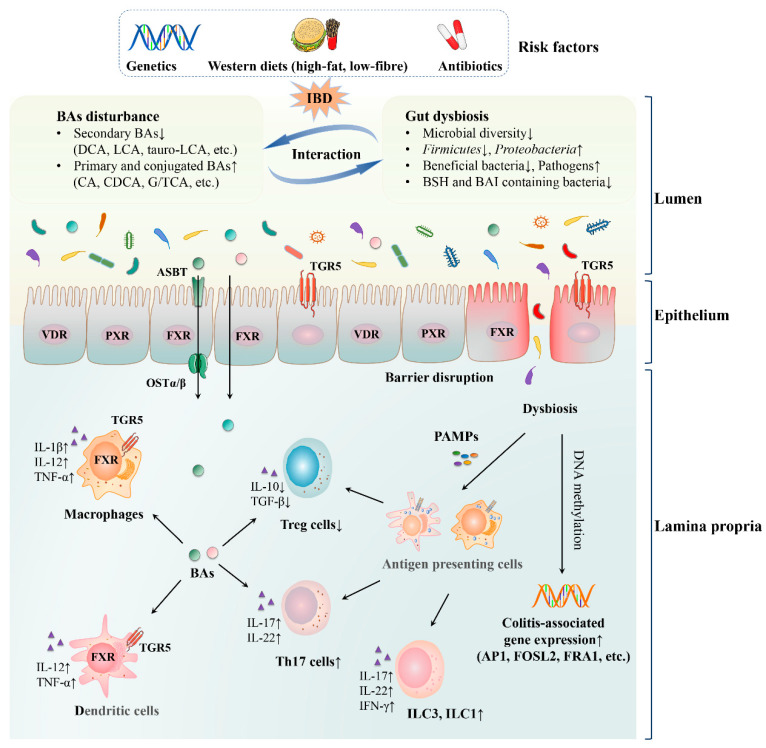
Bile acid–gut microbiota axis in IBD. Abbreviations: DCA, deoxycholic acid; LCA, lithocholic acid; CA, cholic acid; CDCA, chenodeoxycholic acid; GCA, glycocholic acid; TCA, taurocholic acid; FXR, farnesoid X receptor; TGR5, Takeda G-protein receptor 5; PXR, pregnane X receptor; VDR, vitamin D receptor; BSH, bile salt hydrolase; BAI, bile acid-inducible enzymes; IBD, inflammatory bowel disease; PAMPs, pathogen-associated molecular patterns; ASBT, apical sodium-dependent bile acid transporter; OSTα/β, organic solute transporter subunit α/β; Th17 cells, T helper 17 cells; Treg cells, regulatory T cells; ILC3, group 3 innate lymphoid cells; ILC1, group 1 innate lymphoid cells; IL-1β, interleukin-1β; IL-10, interleukin-10; IL-12, interleukin-12; IL-17, interleukin-17; IL-22, interleukin-22; TNF-α, tumor necrosis factor-α; TGF-β, transforming growth factor-β; IFN-γ, interferon-γ.

**Table 1 nutrients-13-03143-t001:** Changes in Gut Microbiota and Bile Acids Profile in Patients with Inflammatory Bowel Disease.

Publication	Patients	Samples and Methods	Major Findings
Duboc et al., 2013 [9]	12 with CD, 30 with UC and 29 HCs	Fecal samples (real-time qPCR and HPLC) Serum samples (HPLC)	*Firmicutes* (e.g., *Clostridium leptum* and *Faecalibacterium prausntizii*) was reduced and *Escherichia coli* was enriched in IBD. Microbial deconjugation, transformation and desulphation capability were depleted in IBD. Fecal conjugated and 3-OH-sulfated BAs were increased, whilst serum and fecal secondary BAs were reduced in IBD.
Sinha et al., 2020 [10]	17 with UC and seven with FAP	Stool samples (metagenomic sequencing and metabolomic analysis)	α-Diversity was reduced and phylum *Firmicutes* and *Ruminococcaceae* (in *Clostridium* cluster XIVa) were decreased in UC. Secondary BAs (LCA and DCA) were remarkably reduced and CDCA was remarkably increased in UC. CA levels were also increased but not significant in UC.
Franzosa et al., 2019 [15]	88 with CD, 76 with UC and 56 non-IBD	Stool samples (metagenomic sequencing and metabolomic analysis)	*Roseburia hominis* and *Ruminococcus obeum* were reduced in IBD. *Ruminococcus gnavus* and *Escherichia coli* were increased in CD. Cholate and CDCA were increased in IBD, whilst secondary BAs (LCA and DCA) were decreased in CD.
Jacobs et al., 2016 [16]	26 with CD, 10 with UC and 54 healthy first-degree relatives	Stool samples (16S rRNA sequencing and HPLC)	Microbial diversity was reduced in CD. *Bifidibacterium adolescentis* and *Parabacteroides distasonis* were enriched, whilst *Faecalibacterium prausntizii* and *Bacteroides fragilis* were reduced in CD. CA, 7-keto-DCA, CDCA sulphate and 3-sulfo-DCA were increased in CD.
Lloyd-Price et al., 2019 [18]	67 with CD, 38 with UC and 27 non-IBDs	Stool samples (metagenomic sequencing and metabolomic analysis)	α-Diversity was decreased and *Faecalibacterium prausnitzii* and *Roseburia hominis* were depleted, whilst *Escherichia coli* was enriched in CD. Primary BA cholate and conjugated BAs (CA, GCA, TCA and GCDCA) were increased, whilst secondary BAs (LCA and DCA) were reduced in IBD.
Wang et al., 2021 [19]	29 pediatric patients with CD and 20 HCs	Fecal samples (16S rRNA sequencing and UPLC-MS)	No significant difference was found in α-diversity between CD and HC. The genera *Bifidobacteria* and *Clostridium* (clusters IV and XI) were decreased in CD. CD showed an increased level of conjugated and primary BAs and a decreased level of unconjugated and secondary BAs (DCA, LCA and hyodeoxycholic acid).
Weng et al., 2019 [37]	173 with CD, 107 with UC and 42 HCs	Fecal samples (metagenomic sequencing and metabolomic analysis) Mucosal biopsy samples (16S rRNA sequencing)	α-Diversity was remarkably reduced in CD. *Enterococcus* and *Hydrogenophilus* were enriched in fecal and mucosal samples of IBD. *Proteobacteria* was enriched in mucosal samples of IBD. LCA, CDCA and tauro-LCA were remarkably decreased in IBD.
Murakami et al., 2018 [38]	Six with CD, six with UC and 26 HCs	Fecal samples (T-RFLP analysis and HPLC) Serum samples (HPLC)	The proportion of fecal *Clostridium* cluster XIVa was remarkably reduced in IBD. Fecal and serum DCA/(DCA + CA) in IBD were reduced compared with those in healthy subjects.
Diederen et al., 2020 [39]	43 pediatric patients with CD and 18 HCs	Fecal samples (16S rRNA sequencing and HPLC)	OTU richness was reduced in CD but the diversity did not remarkably differ between CD and HC. *Eubacterium rectale*, *Bifidobacterium longum* and *Ruminococcus bromii* were increased, whilst *Escherichia coli* was decreased in CD. The relative concentration of primary BAs was increased in CD.
Yang et al., 2021 [22]	32 patients with UC and 23 HCs	Fecal samples (16S rRNA sequencing and UPLC-MS)	*Faecalibacterium*, *Roseburia*, *Butyricicoccus* and *Clostridium* were reduced, whilst *Enterobacteriaceae*, *Enterobacteriales*, and *Escherichia_Shigella* were enriched in UC. The secondary BAs, such as LCA, DCA and tauro-LCA were decreased significantly, whilst primary and conjugated BAs (CA, TCA and G/TCDCA) were increased in UC.

Abbreviations: HPLC, high-performance liquid chromatography; T-RFLP, terminal restriction fragment length polymorphism analysis; UPLC-MS, ultraperformance liquid chromatography coupled with mass spectrometry; OTU, operational taxonomic unit; IBD, inflammatory bowel disease; UC, ulcerative colitis; CD, Crohn’s disease; HC, healthy controls; FAP, familial adenomatous polyposis; BA, bile acid; CA, cholic acid; CDCA, chenodeoxycholic acid; DCA, deoxycholic acid; LCA, lithocholic acid; GCA, glycocholic acid; TCA, taurocholic acid; GCDCA, glycochenodeoxycholic acid.

**Table 2 nutrients-13-03143-t002:** Therapy targeting bile acid–gut microbiota axis for inflammatory bowel disease.

Publication	Subjects	Treatment	Samples and Methods	Major Findings
Diederen et al., 2020 [39]	43 pediatric patients with CD and 18 healthy controls	EEN for 6 weeks followed by 2 weeks of EEN tapering	Fecal samples (16S rRNA sequencing and HPLC)	EEN decreased the microbiota diversity and reduced trimethylamine and cadaverine towards control levels. Reduced microbial metabolism of BAs in CD was partially normalized during EEN.
Paramsothy et al., 2019 [88]	81 patients with active UC	FMT or placebo colonoscopic infusion, followed by enemas 5 days per week for 8 weeks	Fecal samples (metagenomic and metabolomic analysis) Colonic biopsy samples (16S rRNA gene and transcript sequencing)	Microbial diversity was increased and the composition was altered after FMT. The patients in remission had enriched *Eubacterium hallii* and *Roseburia inulivorans*. Mucosal microbiota showed increased α-diversity after FMT. Patients in remission had increased levels of secondary BA (dehydrolithocholate).
Wang et al., 2021 [19]	29 pediatric CDs	Infliximab infusion for 3–6 times	Fecal samples (16S rRNA sequencing and UPLC-MS)	The abundances of *Blautia*, *Clostridium IV*, *Collinsella*, *Eubacterium* and *Ruminococcus* were increased after treatment. The ratios of unconjugated/conjugated BAs and secondary/primary BAs were elevated.
Wang et al., 2019 [89]	Canine model of chronic inflammatory enteropathy	Hydrolyzed protein diet for 6 weeks	Fecal samples (metagenomic and metabolomic analysis)	Gut microbiota was restored, as marked by reduced pathogens and increased *Clostridium hiranonis* after treatment. The levels of secondary BAs (LCA and DCA) were increased after treatment.
Ke et al., 2020 [90]	Mice with DSS-induced chronic colitis	Fucose gavage for 57 days	Ileal tissue lysates and colonic feces (16S rRNA sequencing and UPLC-MS)	Fucose increased α-diversity and reversed the decreased ratio of *Firmicutes* to *Bacteroidetes*. The level of tauro-β-MCA and TCA was decreased and the abnormal ratio of conjugated/unconjugated BAs was restored after treatment.
Jia et al., 2020 [91]	Mice with DSS-induced acute colitis	Oral total alkaloids of Sophora alopecuroides L. for 7 days	Cecum content (16S rDNA gene sequencing) Liver, bile, serum, cecum content and colon samples (UPLC-MS)	The abundance of *Firmicutes* was increased, whereas that of *Bacteroidetes* was decreased after treatment. The elevated MCAs and CA were restored after treatment.
Bossche et al., 2017 [12]	Mice with DSS-induced acute colitis	Daily gavage of UDCA, TUDCA, GUDCA or placebo for 10 days	Fecal samples (16S rRNA sequencing and HPLC)	The ratio of *Firmicutes/Bacteroidetes* was normalized and the abundance of *Clostridium* cluster XIVa and *Akkermansia muciniphila* was increased. The concentrations of UDCA, TUDCA and LCA were remarkably elevated after treatments.

Abbreviations: FMT, fecal microbiota transplantation; EEN, exclusive enteral nutrition; IBD, inflammatory bowel disease; UC, ulcerative colitis; CD, Crohn’s disease; DSS, dextran sulfate sodium; BA, bile acid; HPLC, high-performance liquid chromatography; UPLC-MS, ultraperformance liquid chromatography coupled with mass spectrometry; MCA, muricholic acids; CA, cholic acid; TCA, taurocholic acid; UDCA, ursodeoxycholic acid; TUDCA, tauroursodeoxycholic acid; GUDCA, glycoursodeoxycholic acid; DCA, deoxycholic acid; LCA, lithocholic acid.

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
