# Peer review of "Bile Acid–Gut Microbiota Axis in Inflammatory Bowel Disease: From Bench to Bedside"

_nutrients, 2021, doi:10.3390/nu13093143_

Round 1

Reviewer 1 Report

In the manuscript ID-nutrients-1344653 titled “Bile acid–gut microbiota axis in inflammatory bowel disease: from bench to bedside” by Min Yang and colleagues, the authors have reported that the concentration of primary and conjugated bile acids is elevated at the expense of secondary bile acids in IBD. In turn, bile acids could modulate the microbial community. Gut dysbiosis and disturbed bile acids impair the gut barrier and immunity. Several therapies, such as diets, probiotics, prebiotics, engineered bacteria, fecal microbiota transplantation, and ursodeoxycholic acid, may alleviate IBD by restoring gut microbiota and bile acids. Thus, the bile acid–gut microbiota axis is closely connected with IBD pathogenesis. Regulation of this axis may be a novel option for treating IBD. I have few concerns regarding the present manuscript.

-Why does the title has this, BA-microbiota axis in IBD.

-The manuscript is well-organized and the authors have reported important information, maybe the prebiotics and probiotics section need more information.

-Microbial signature information in IBD and then the bile acid effects relationship might state. Maybe an IBD figure with microbiota and BA is required.

Reviewer 2 Report

Very nice and neat review of a very important and contemporary topic. 

Author Response

This manuscript is a resubmission of an earlier submission. The following is a list of the peer review reports and author responses from that submission.